# Changes in Air-Pollution-Related Information-Seeking Behaviour during the COVID-19 Pandemic in Poland

**DOI:** 10.3390/ijerph19095613

**Published:** 2022-05-05

**Authors:** Wojciech Nazar, Marek Niedoszytko

**Affiliations:** 1Faculty of Medicine, Medical University of Gdańsk, Marii Skłodowskiej-Curie 3a, 80-210 Gdansk, Poland; 2Department of Allergology, Medical University of Gdańsk, Smoluchowskiego 17, 80-214 Gdansk, Poland; mnied@gumed.edu.pl

**Keywords:** air pollution, information-seeking behaviour, Poland, machine learning, COVID-19

## Abstract

Low air quality in Poland is a problem of particularly high urgency. Therefore, Poles must be aware of air quality levels, also during the COVID-19 pandemic. The study aimed to compare air-pollution-related information-seeking behaviour between the pre- and intra-pandemic periods as well as between the actual and theoretical machine-learning-forecasted intra-pandemic models. Google Trends search volumes (GTSVs) in Poland for air-pollution-related keywords were collected between January 2016 and January 2022. To investigate the changes that would have occurred without the outbreak of the pandemic, Seasonal Autoregressive Integrated Moving Average (SARIMA) machine-learning models were trained. Approximately 4,500,000 search queries were analysed. Between pre- and intra-pandemic periods, weighted mean GTSVs changed by −39.0%. When the actual intra-pandemic weighted mean GTSVs were compared to the intra-pandemic forecasts, the actual values were lower by −16.5% (SARIMA’s error = 6.2%). Compared to the pre-pandemic period, in the intra-pandemic period, the number of search queries containing keywords connected with air pollution decreased. Moreover, the COVID-19 pandemic might have facilitated the decrease. Possible causes include an attention shift towards everyday problems connected to the pandemic, worse mental health status and lower outdoor exposure that might have resulted in a lower intensity of non-pandemic-related active information-seeking behaviour.

## 1. Introduction

### 1.1. Air Pollution in Poland

It is estimated that, in 2019, in the European Union Member States over 360,000 premature deaths were attributable to exposure to fine particulate matter, nitrogen dioxide and ozone. This makes air pollution the single largest environmental health risk in Europe [1]. Air pollution in Poland is a problem of particularly high urgency. Even though the air pollution levels in large polish urban areas over the last decade are decreasing, World Bank Group states that 36 of the 50 most polluted cities in the European Union are in Poland [2,3].

It was found that high air pollution levels in Poland contribute significantly to higher general and respiratory disease mortality rates [4,5]. According to a large international study, certain regions of Poland have some of the highest estimated premature mortality burdens attributable to air pollution in Europe [6]. An increase in particulate matter concentration is also responsible for a higher number of respiratory disease hospitalizations and outpatient visits [7].

Moreover, Polish citizens living in polluted areas have worse spirometry outcomes [8,9]. Polish children that were pre- and postnatally exposed to high particulate matter concentrations have impaired lung function that does not compensate over time [10]. Chronic exposure to high air pollution levels seems to increase the prevalence of lung cancer in adults and the prevalence of respiratory diseases in children [11,12]. The quality of life of asthmatic patients is also decreased when the air pollution is higher [13].

In the years 2016–2019, the 2005 World Health Organisation (WHO) guidelines on the annual air quality levels were not met in Poland in any voivodeship (province) nor in any voivodeship’s capital city [14]. In 2021, the WHO reviewed its air quality guidelines and significantly reduced the maximum recommended mean annual concentrations of major air pollutants, namely particulate matter and nitrogen oxide [15]. This further emphasised the health risks attributable to breathing low-quality air. In the years 2016–2019, the air quality was worse in the south of Poland and during the winter season [14].

In such a context, it is vital that people exposed to highly polluted air are aware of the changes in air quality and respond to it. It was found that, during the 2016–2019 period, air pollution information-seeking behaviour was more frequent in the southern part of Poland and during the winter season and that it corresponded to the seasonal and spatial distribution of higher air pollution levels [14]. Therefore, before the COVID-19 pandemic, Poles seemed to respond to elevated air pollution levels in expected and apparently rational ways as far as information-seeking was concerned.

### 1.2. Impact of the COVID-19 Pandemic

Following the COVID-19 pandemic outbreak, many aspects of everyday life changed. Due to lockdowns, people began to spend more time at home [16]. Mental health indicators decreased worldwide, as confirmed by international and Polish studies [17,18,19]. Moreover, many people and societies worldwide excessively focused on COVID-19 to the detriment of other issues. For example, in the US, about 40% of adults delayed or avoided medical care during the initial COVID-19 outbreak [20]. In a nationwide cohort study in Denmark, it was noted that hospital admissions for all major non-COVID-19 disease groups decreased during the national lockdowns [21].

In a national Polish survey, 37% of oncological patients experienced a postponement of their treatment, while appointments with clinical oncologists/radiotherapists were postponed in over 50% of cases [22]. During the national lockdown in April–May 2020, the number of cancer-screening mammography tests decreased by over 90%, while the number of Pap smears decreased by more than 85% [23]. Therefore, the size of the national secondary prevention programs for breast and cervical cancer was greatly reduced [23]. 

These changes may have resulted from the initial fears and behavioural changes in the Polish population related to the pandemic. People started to frequently stay at home and were worried about their families, the possible economic crisis and potential quarantine [16]. On the other hand, during the lockdowns, such behavioural changes resulted in local air pollution reduction [24,25].

Further on, the spread of COVID-19 itself appears to have been facilitated by increased air pollution levels. Such relationships were noticed not only in Poland but also in other studies conducted in China, Chile, India, Italy, Russia and Qatar [26,27,28,29].

Thus, it is important to investigate the changes in Poles’ air pollution information-seeking behaviour during the COVID-19 pandemic—even more so as there is a possible causative link between higher air pollution levels and higher COVID-19 spread. Additionally, it is of great interest to determine if Poles’ interest in air pollution during the intra-pandemic period changed significantly in comparison to the theoretically forecasted (modelled) interest in air pollution for if there was no outbreak of COVID-19.

### 1.3. Aims

The primary aim of the study was to compare changes in the intensity of air-pollution-related information-seeking behaviour between the pre- and intra-pandemic periods as well as between the actual behaviour and theoretical machine learning-forecasted intra-pandemic models.

The secondary objective included the analysis of correlations between intra-pandemic air-pollution-related GTSVs and the number of confirmed COVID-19 cases and deaths in Poland and Europe.

## 2. Materials and Methods

### 2.1. Keyword Search Tools

Google Trends is a free keyword search tool provided by Google [30]. It shows the weekly interest in a given keyword using search volume figures. The weekly search volume shows the relative interest in a given keyword during a selected time frame. Its value ranges between 0 and 100. A value of 100 is ascribed to the day during which the keyword was searched the greatest number of times. A value of 0 corresponds to a relative quantity of search queries equal to less than 1% of its greatest popularity over the selected time frame. Thus, data on the relative interest in a given keyword over the investigated period can be collected. 

However, the annual absolute search volumes for selected keywords are not reported by Google Trends. Thus, a KWFinder, another keyword research tool, developed by Mangools, was used to find the general search volumes based on absolute values [31].

Searches for all keywords were performed on 30 January 2022.

### 2.2. Identification of Pre- and Intra-Pandemic Periods in Poland

Google Trends was used to identify the first mentions of the “Coronavirus” (“Koronawirus” in Polish) and “COVID-19” (COVID-19) in Poland. The keyword search was performed for Poland and in Polish. From 1 January 2016, an improved version of Google Trends was available [30]. Thus, this day was used as the initial date during the Google Trends search. 

The first measurable mentions of the pandemic-related terms occurred at the end of January 2020 (19 January 2020–25 January 2020). This correlates with the first detected cases of COVID-19 in Europe (France on 24 January 2020) [32]. 

To maintain the weekly cycle of the data, 26 January 2020 (the first Sunday after the COVID-19 pandemic outbreak in Europe) was set as the initial day of the intra-pandemic period. It was possible to analyse two whole years of the pandemic, and therefore 29 January 2022 was set as the last day of the intra-pandemic period. The dataset included 735 calendar days (105 weeks). A corresponding seasonally and weekly adjusted pre-pandemic dataset that included the period between 24 January 2016 and 25 January 2020 was created (1463 calendar days; 209 weeks).

### 2.3. Air-Pollution-Related Keyword Search

First, keywords related to air pollution were proposed by the authors. Subsequently, to generate data, each keyword was separately entered into Google Trends. The search period was set between 25 January 2016 and 29 January 2022. 

Data on the national level for the following eight keywords were available: “smog” (smog), “air pollution” (zanieczyszczenie powietrza), “air quality” (jakość powietrza), “air quality index” (indeks jakości powietrza), “air purity” (czystość powietrza), “PM_10_” (PM_10_) and “PM_2.5_” (PM_2.5_).

### 2.4. Weight Calculation

For each of the keywords, GTSV basic exploratory analyses were completed to identify seasonal and annual trends in the studied time series dataset (2198 days; 314 weeks).

Additionally, arithmetic and weighted keywords’ weekly GTSVs were calculated. The weights were calculated in the following way:First, the absolute monthly search volumes (AMSVs) for the keywords were determined using KWFinder. KWFinder approximates AMSV based on annual interest in the given keyword. Therefore, by multiplying AMSV by 12, the absolute annual search volume was calculated.Next, the sum of GTSV for the first year (52 weeks; 364 days) and for the whole studied period were calculated. Next, the 6-year sum of GTSV was divided by the GTSV figure for 2021.Subsequently, the annual absolute search volume for each keyword was multiplied by the keyword’s 6-year coefficient obtained in step 2 above. In this way, estimated 6-year absolute search volumes were calculated. These were eventually used as weights to calculate the weighted weekly GTSV.

### 2.5. Intra-Pandemic Google Trends Search Volume Forecasting

As most of the keywords’ weekly GTSVs showed both seasonal as well as annual trends, a Seasonal Autoregressive Integrated Moving Average (SARIMA) machine-learning model for each keyword as well as for weighted and arithmetic keywords’ GTSV averages were created (nine models in total). The SARIMA model supports univariate time series data input with a seasonal component, which made it suitable for analysis of changes in the GTSVs for GTSV forecasting. 

As the number of weeks varies between years, to prevent a “frameshift” between given years, the mean monthly GTSVs for the years 2016–2019 were used as input data. 

The SARIMA model is specified as: (1)SARIMA(p,d,q)×(P,D,Q)m

The “p”, ”d” and “q” hyperparameters describe the annual trends, while the “P”, ”D”, “Q” and “m” hyperparameters describe the seasonal trends. The “m” hyperparameter was set to 12, which corresponds to the number of time steps for a single seasonal period (12 months).

To search for “d” and “D” values, an augmented Dickey–Fuller (ADF) test was performed to check for the stationarity of the data. If the alternative hypothesis was accepted, the data were taken as stationary, and the “d” and “D” hyperparameters were set to “0”. If the null hypothesis was accepted, the data underwent first differencing, and the ADF test was repeated. If then the alternative hypothesis was accepted, the “d” and “D” hyperparameters were set to “1”. 

Next, the “p”, “q”, “P” and “Q” hyperparameters were grid searched. The range for grid search was estimated using plots of autocorrelation and partial autocorrelation functions. The Akaike’s Information Criterion (AIC) was used to find the combination of all hyperparameters with the best fit. A model with the lowest AIC was chosen. To evaluate the goodness of fit, the mean absolute percentage errors (MAPEs) between the modelled and actual GTSV for the pre-pandemic period were calculated.

Based on the four-year model, a two-year prediction of the weekly GTSVs for a given keyword for the intra-pandemic period (2020–2021) was performed. Next, the data forecasted with SARIMA and the real-world intra-pandemic GTSVs for the intra-pandemic period were compared statistically. 

Such an approach provides a methodological advantage over the simple comparison of pre- and intra-pandemic GTSVs, as the forecasts made with the use of the SARIMA models represent hypothetical changes in the seasonal and annual trends that would have potentially occurred if the was no outbreak of the COVID-19 pandemic. Therefore, it is possible to find out if the COVID-19 pandemic could be responsible for the changes in air-pollution-related information-seeking behaviour among Poles.

### 2.6. Obtaining and Analysis of the COVID-19 Pandemic Dataset

A number of daily new confirmed COVID-19 cases and new deaths in Poland and in Europe were acquired from the Our World in Data database [33]. To adjust the COVID-19 pandemic data to the GTSV data, monthly averages of the metrics were calculated. Additionally, monthly GTSVs for the “COVID-19” keyword in Poland for the intra-pandemic period were collected.

Next, for this analysis only, for intra-pandemic GTSV data as well as for the COVID-19 pandemic dataset, the maximum values of each metric were identified. Subsequently, for each data point, a percentage value relative to the maximum observed value for a given metric was calculated. Thus, the relative value of all data points was expressed as a percentage, which simplified the visual analysis.

### 2.7. Statistical Analysis

Real-world intra-pandemic GTSVs and real-world pre-pandemic GTSVs were summarized with the use of a box plot based on the weekly GTSV values. For the comparison between real-world and forecasted intra-pandemic GTSVs, the data were summarized with the use of a box plot based on the mean monthly GTSV values.

To investigate changes in GTSVs between the pre- and intra-pandemic periods, the real-world intra-pandemic GTSVs were separately compared to the real-world pre-pandemic GTSVs and the forecasted intra-pandemic GTSVs and were eventually analysed with the use of Student’s *t*-test or the nonparametric Wilcoxon rank-sum test (Mann–Whitney U test). 

The null hypothesis stated that there was no statistically significant difference between the compared GTSVs. The threshold of statistical significance was set at *p* < 0.05. In addition to that, for the COVID-19 pandemic dataset, Pearson product–moment correlation coefficients between the mean percentage of new cases, new deaths and GTSVs were calculated. All analyses were performed in Python 3.8.8 and with the use of additional packages available in Anaconda distribution 4.10.3 (Anaconda.org, New York, NY, USA).

## 3. Results

In total, about 4,500,000 queries were analysed (Table 1). “Air pollution”, “smog” and “air quality” were keywords with the greatest number of searches (about 1,500,000, 2,200,000 and 590,000 searches, respectively). On average, about 2068 searches occurred daily. 

We found that, for most of the keywords, the weekly intra-pandemic GTSV decreased in comparison to the weekly pre-pandemic GTSV (Figure 1 and Table 2). The greatest decrease was observed for queries regarding “air pollution” and “smog” (−53.5% and −51.0%, respectively). For these keywords, a statistically significant difference between the pre- and intra-pandemic periods was observed. Moreover, for all keywords, a reduction of the interquartile range was noted. Overall, the arithmetic and weighted mean GTSVs changed by −18.7% and −39.0%, respectively. However, the differences were not statistically significant.

Overall, an adequate visual fit was observed for all models except the “PM_10_” model (Figure 2). Most of the SARIMA models reported the best fit with hyperparameters set to (1,0,1) × (1,0,1)12 (Table 3). Large discrepancies in MAPE between keywords were noted (from 2.2% for “air quality index” to 44.3% for “PM_2.5_”). For the arithmetic and weighted mean GTSV models, the MAPE values were fairly small (10.2% and 6.2%, respectively).

Except for the “PM_10_” and “PM_2.5_” terms, a reduction in the real-world intra-pandemic mean monthly GTSV in comparison to the forecasted mean monthly GTSV was observed (Figure 3, Table 3). The greatest decrease was observed for “air purity”,” air quality” and “air pollution” (−56.9%, −41.3% and −33.0%, respectively). Statistically significant differences between the forecasted and real-world data were noted for “air purity” and “PM_2.5_” only. For arithmetic means, only a slight decrease was observed (−3.3%), while for weighted means, a moderate decrease of −16.5% was noted.

The correlations between the real-world and forecasted GTSVs as well as between weighted and arithmetic means of GTSVs were positive and strong (ρ > 0.85). We also found that the changes in the intra-pandemic GTSVs regarding keywords related to air pollution were weakly to moderately positively correlated with the number of COVID-19 cases and deaths in Poland and Europe (Figure 4 and Figure 5). The strongest correlations were observed between the following metrics: Real-world GTSV arithmetic mean and new COVID-19 deaths in Europe (ρ = 0.44).Forecasted GTSV weighted mean and new COVID-19 cases in Europe (ρ = 0.41).Real-world GTSV arithmetic mean and new COVID-19 deaths in Poland (ρ = 0.40).Real-world GTSV weighted mean and new COVID-19 deaths in Europe (ρ = 0.39).

There was no correlation between the changes in the intra-pandemic GTSVs regarding keywords related to air pollution and the “COVID-19” keyword (ρ from 0.055 to 0.11). On the other hand, searching for the “COVID-19” keyword was strongly correlated with the number of new COVID-19 cases and deaths in Poland (ρ = 0.82 and ρ = 0.73, respectively).

## 4. Discussion

### 4.1. Interpretation of the Results

Over 4,500,000 search queries were analysed, which gives about 2068 queries per day (Table 1). This can be considered a relatively low number considering the population of Poland (approximately 38,000,000) [34]. In addition to that, very large differences in the number of search queries between keywords were observed, from about 50,000 for “PM_2.5_” to over 2,000,000 for “smog”. Thus, it made sense to calculate the weight for each keyword and to compute the weighted mean GTSV for all keywords combined.

Compared to the pre-pandemic period, during the intra-pandemic period, a large decrease in the number of search queries regarding keywords connected with air pollution was noticed (Table 2, Figure 2). For the two most commonly searched keywords, “air pollution” and “smog”, accounting for about 80% of all analysed search queries, −53.5% and −51.0% decreases, respectively, were observed. Both of these differences were statistically significant. A large change was also observed for the weighted mean GTSV, amounting to −39.0%; however, the difference was not statistically significant. It can be thus concluded that, during the intra-pandemic period, the social awareness of air pollution decreased remarkably.

It was important to determine whether the noted decreases were solely due to natural changes that would have occurred without the COVID-19 pandemic outbreak or if the decreases were caused by the attention shift to the ongoing lockdowns and changes in everyday life of societies worldwide. To investigate this issue, nine SARIMA machine-learning models were developed. As the models were trained on the data from the pre-pandemic period, it was possible to model a theoretical, natural course of the changes in the interest in the selected keywords. The model was then compared to the actual changes that occurred during the COVID-19 pandemic.

According to the visual analysis of the models, all of them except for the “PM_10_” keyword model fitted well with the corresponding training datasets (Figure 3). Most likely, the causes of the poor fit of the “PM_10_” model are the relatively low popularity of the keyword as well as possible interference of this search term with longer or less-precise search queries.

Further on, all models with accurate fit presented a clear seasonal pattern of GTSV changes (higher in the winter and lower in the summer) as described in previous studies [14]. Additionally, strong positive correlations between the real-world and forecasted arithmetic/weighted mean GTSVs were identified (ρ > 0.85, Figure 5), which confirms the good fit of the models. The MAPEs for the models regarding arithmetic and weighted GTSV analysis were fairly good (10.2% and 6.2%, respectively; Table 3). These prove a good fit for the trained machine-learning models. In the case of the “PM_10_” keyword, the pattern of its GTSV changes is rather irregular, and therefore it was difficult to fit an accurate model describing these changes. Thus, in the case of this keyword, the model can be found to be inaccurate in predicting future GTSV changes.

When the actual intra-pandemic GTSVs were compared to the forecasted GTSVs, for the two most common keywords, “air pollution” and “smog”, moderately large decreases of −33.0% and −19.7%, respectively, in the real-world values were observed (Table 3). However, for “air purity” and “air quality” search terms, a reduction of actual versus forecasted search intensity was −56.9% and −41.3%, respectively. On the other hand, for the “PM_2.5_” term the actual GTSVs were over 130% higher than the ones projected with the use of the machine-learning model. Overall, the actual weighted mean GTSVs were on average −16.5% lower than the forecasted weighted mean GTSVs. The calculated error of the model was 6.2%. Thus, based on this data, it can be concluded that to some extent the outbreak of the COVID-19 pandemic might have facilitated the ongoing decrease in the intensity of air-pollution-related information-seeking behaviour in Poland.

On the other hand, we found that the changes in the intra-pandemic GTSVs for keywords related to air pollution were weakly to moderately positively correlated with the number of COVID-19 cases and deaths in Poland and in Europe (ρ~0.40) (Figure 4 and Figure 5). This suggests that an increase in COVID-19 cases and/or deaths would result in an increase in air-pollution-related information-seeking behaviours, which is possible as the fear of serious threat of the COVID-19 infection might have motivated people to analyse other factors that could have affected the respiratory system at that time. 

On the other hand, if the correlations supported the observed decreases in the mean GTSVs (Table 3), negative correlation coefficients would be expected. In addition to that, we noted that the intensity of “COVID-19” keyword search queries was certainly not correlated with the changes in both the forecasted and actual mean GTSVs (ρ from 0.055 to 0.11). Therefore, based on the data provided by the correlation coefficients, the changes in air-pollution-related information-seeking behaviours appeared to be independent of the events of the COVID-19 pandemic.

Rousseau et al. reported that, according to the Google Trends data collected for the largest countries in Europe, the relative interest in air pollution at the beginning of the pandemic did not change [35]. However, a short period was investigated (from 1 January 2019 until 11 May 2020), and the investigation encompassed only one keyword (“air pollution”) in twelve European countries. Therefore, even though, in the short-term and on the European-scale, the change in information-seeking behaviour may not be clearly visible, it does not eliminate the possibility of long-term country-scale changes. 

Altogether, the evidence collected in our study does not allow for determining if the decrease in air pollution information-seeking behaviour of Poles was facilitated by the COVID-19 pandemic. On the one hand, the difference in the actual intra-pandemic weighted mean GTSV versus the forecasted intra-pandemic weighted mean GTSV was −16.5% with a MAPE of 6.2%. Based on a simple comparison, it appears that the decrease in the actual GTSV was greater than the model error. Nonetheless, for the comparison of the actual versus forecasted arithmetic mean GTSVs, the values were −3.3% and 10.2%, respectively. 

Based on correlation coefficients, the events of the pandemic appear to be independent of the changes in relative intensity of air-pollution-related information-seeking behaviour. Therefore, the effect of the COVID-19 pandemic outbreak remains unclear. However, as the differences in the absolute keyword search volumes were large (Table 1), the model based on weighted means appears to express the real-world conditions more accurately than the simpler arithmetic mean model. 

As the correlations were positive and weak-to-moderate, there was likely no causal link between the social awareness of air pollution and changing metrics of the COVID-19 pandemic, or the changes were too small to be measurable. It is possible that the outbreak of the pandemic might have facilitated a slight decrease in air-pollution-related information-seeking behaviour; however, the difference, if it occurred, was fairly small.

### 4.2. Potential Explanations of the Results

A potential explanation of the possible pandemic-facilitated decrease in GTSV may be found in the dynamic changes in behaviours, fears and practices of people that occurred after the COVID-19 pandemic outbreak. In a study regarding the knowledge and perception of the COVID-19 pandemic conducted in two waves in March–April 2020 in Poland, we observed that the respondents were progressively more scared of the pandemic and economic crisis and were worried about their families [16]. 

These concerns may have caused an attention shift towards the outbreak of COVID-19 and the need for constant follow-up regarding the rapidly changing lockdown restriction guidelines. Indeed, according to our current analysis, the “COVID-19” search term was well correlated with the changes in the number of new cases of COVID-19 and deaths due to COVID-19 in Poland (ρ = 0.82 and ρ = 0.73, respectively; Figure 4 and Figure 5). Thus, an attention shift among Poles was quite probable. Moreover, during each COVID-19 wave, the information-seeking behaviours related to COVID-19 increased [16]. 

Therefore, the intensity of the possible attention shift likely varied with time during the pandemic. Changes in information-seeking behaviours during the beginning of the COVID-19 pandemic were also observed in the US [36]. Based on Google Trends, Tichakunda et al. reported that changes occurred in the categories regarding care-seeking, news and influence as well as outlook and concerns [36]. Therefore, reduced information seeking concerning needs perceived as less urgent, such as the local air quality index, might have occurred.

Further on, especially during the lockdowns, Poles changed their everyday behaviours and began to stay at home more and wear face coverings [16]. Thus, both the exposure to outdoor air pollution as well as the potential amount of inhaled toxic particles were reduced. This might have resulted in a decrease in the interest in air pollution levels, as the primary reason for information seeking (going outside) was less frequent, and overall the outdoor safety level (mouth and nose coverage) was greater.

Moreover, during the COVID-19 pandemic, Poles also changed other aspects of their everyday life. For example, reductions in time devoted to physical activity, increases in sedentary lifestyle and modifications of dietary behaviours [37,38,39,40] were observed. Thus, people might have addressed their primary needs and attempted to cope with the multi-faceted everyday lifestyle changes, leaving secondary problems, such as the air quality index, out of focus.

In addition, elevated stress levels during the COVID-19 pandemic in Poland resulted in the frequently reported negative impact of the pandemic on mental health; increased intake of alcohol, cigarettes and other stimulants; as well as elevated perceived levels of loneliness or daily life fatigue [17,41,42,43,44,45]. Reduced mental health was observed among all age groups from students to elderly people [17,41,42,43,44,45]. Thus, due to the generally worse well-being status of Poles during the pandemic, the intensity of all other non-pandemic-related active information-seeking behaviours might have been diminished as well.

Last, the problem of low air quality in Poland has been measured for over 20 years [46]. As this issue was highly promoted in the last decade, habituation to air pollution by Poles might have occurred. Recently, similar observations were described for several highly polluted countries [47]. As suggested by the authors of this study, such behaviour may threaten societies’ collaboration towards air pollution reduction. 

This emphasizes the alarming evidence of our study and the constant need for local air pollution awareness promotion, especially in the highly polluted regions of Poland. Fortunately, based on the example of Cracow, a large city located in southern Poland, it is clearly visible that education as well as intensive multilevel political involvement are effective initiatives in increasing the environmental awareness of the local population, which results in a stable decrease in particulate matter concentrations [3].

### 4.3. Future Perspectives

Independently of the cause, in the recent years, a large reduction in the intensity of information-seeking behaviours regarding air pollution in Poland was noted. Therefore, urgent, government-subsidized action is needed to promote social awareness of the local air quality levels. In addition to that, further in-person multicentre studies are needed to investigate if people have actually reduced the intensity of their air-pollution-related information-seeking behaviours. Causes of such behaviour should be identified, and effective strategies to tackle these issues should be developed.

### 4.4. Limitations

The study is subject to methodological limitations. First, the authors used data provided by Google Trends and did not collect the number of given search queries by themselves, which might have introduced bias into the analysis. Moreover, sometimes the GTSVs reported by Google Trends may differ—for example, when the entered search query is embedded in quotation marks in comparison to the situation when the term is entered without them. To reduce this bias, each search term was always entered without quotation marks and without with special characters.

Additionally, only seven keywords related to air pollution were included in the analysis. A larger number of keywords would result in a greater absolute volume of analysed search queries, which would result in more data and more accurate conclusions. However, GTSV data in the selected time frame were available in Poland only for the selected seven keywords.

Further, as language and recent trends evolve, so does the popularity of different search terms. Therefore, it is possible that particular keywords might have been popular at the beginning of the studied period and that their popularity with time decreased, while the general interest in the topic of air pollution did not. To reduce this bias, we analysed seven keywords as well as the mean arithmetic and mean weighted GTSVs. The changes in keyword popularity might have also affected the absolute search volumes estimated with use of KWFinder. Therefore, the uncertainty of the calculated absolute search volumes may be high and difficult to estimate.

Last but not least, machine-learning models do not fit the data in an ideal way. Therefore, the forecasts made with the use of SARIMA models have an error that might have affected the calculated differences between the forecasted and actual intra-pandemic GTSVs. Again, the authors aimed to reduce this bias with the use of several SARIMA models that were calculated for each keyword separately as well as for all keywords combined. Moreover, MAPEs were calculated to estimate the accuracy of the machine-learning models.

## 5. Conclusions

Compared to the pre-pandemic period, during the intra-pandemic period, a large decrease in the number of search queries for keywords connected with air pollution was observed. The COVID-19 pandemic might have facilitated the decrease in the number of keywords searches regarding air pollution awareness in Poland.

Possible causes of reduced air-pollution-related information-seeking behaviours during the pandemic include the attention shift towards the everyday fears and problems regarding the pandemic. Additionally, people might have focused on multi-faceted everyday changes caused by lockdowns, such as reduced physical activity, increased time spent at home or changes in dietary behaviours. Further on, especially during lockdowns, Poles’ exposure to outdoor air pollution was decreased. Moreover, overall reduced mental health and well-being might have resulted in a lower intensity of non-pandemic-related active information-seeking behaviours.

On the other hand, differences between the forecasted and actual data may partially result from the methodological limitations of machine-learning algorithms. Independently of the cause, during recent years, a large reduction in the intensity of information-seeking behaviours regarding air pollution in Poland was noted, and urgent government-subsidized action is needed to promote social awareness of the local air quality levels.

## Figures and Tables

**Figure 1 ijerph-19-05613-f001:**
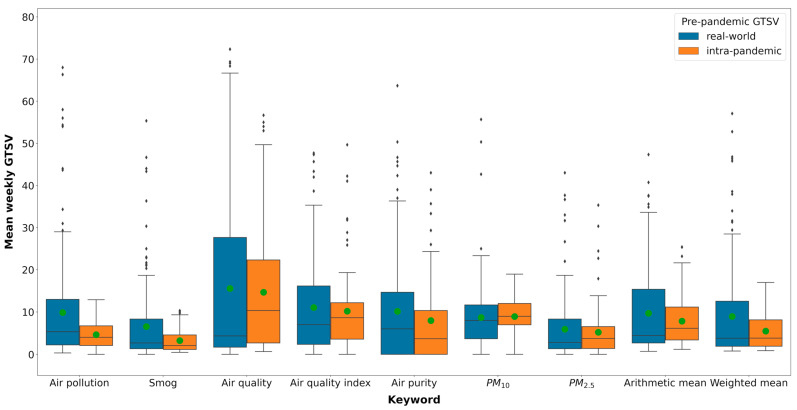
Comparison of the real-world pre- and intra-pandemic GTSVs. Green dots represent the mean values.

**Figure 2 ijerph-19-05613-f002:**
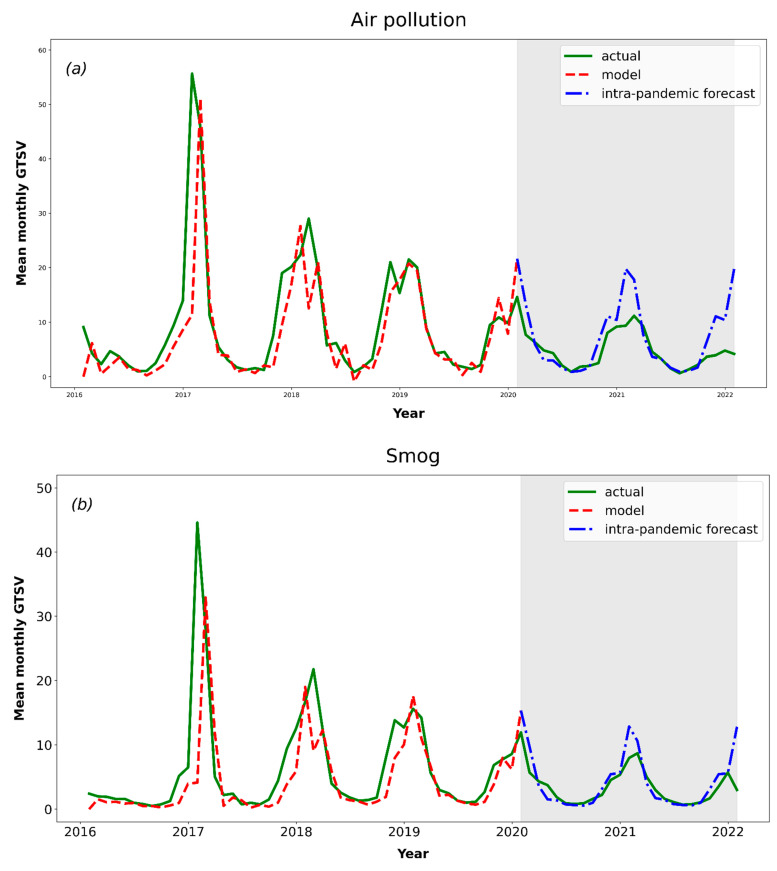
SARIMA models for the analysed keywords: “air pollution” (**a**), “smog” (**b**), “air quality” (**c**), “air quality index” (**d**), “air purity” (**e**), “PM_10_” (**f**), “PM_2.5_” (**g**), arithmetic mean (**h**) and weighted mean (**i**).

**Figure 3 ijerph-19-05613-f003:**
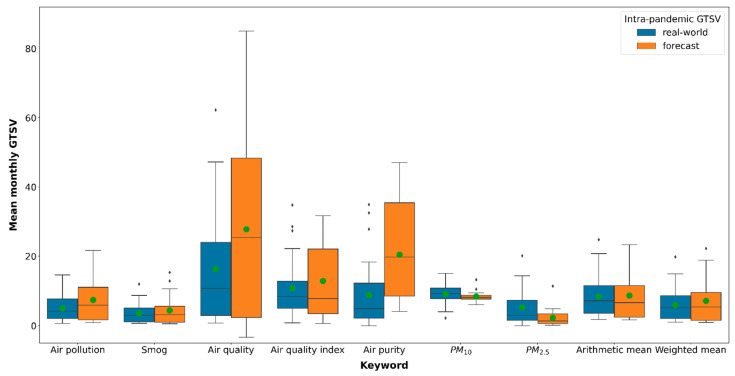
Comparison of the forecasted and real-world intra-pandemic GTSVs. Green dots represent the mean values.

**Figure 4 ijerph-19-05613-f004:**
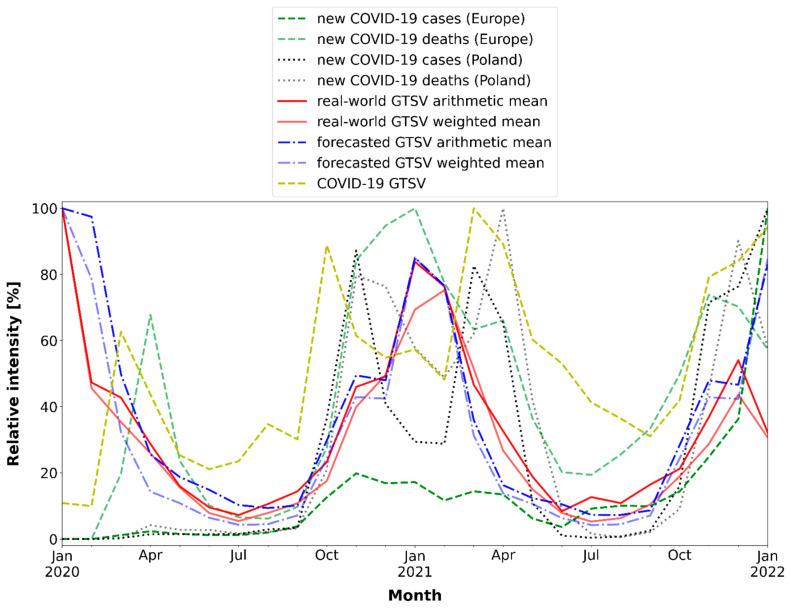
Changes in the COVID-19 pandemic metrics and GTSVs.

**Figure 5 ijerph-19-05613-f005:**
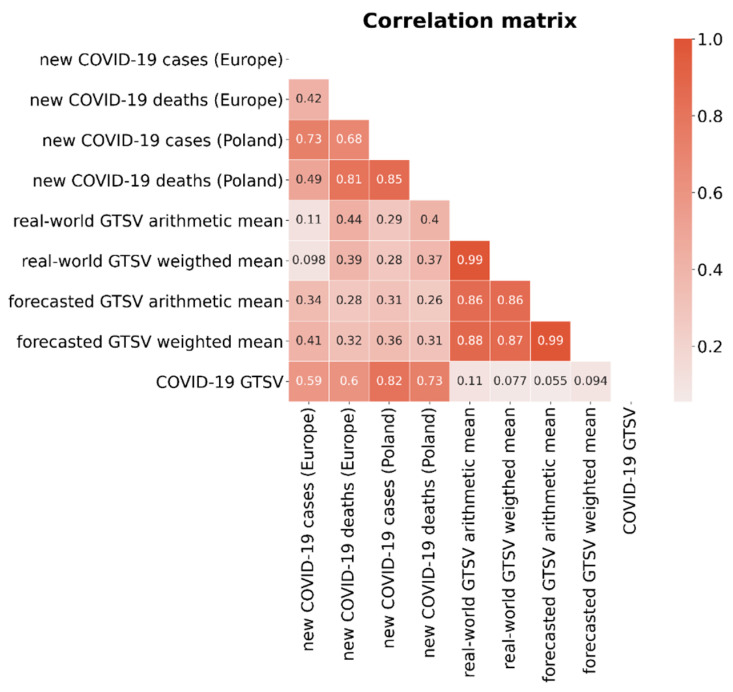
Pearson product–moment correlation coefficient matrix regarding the mean monthly changes in COVID-19 pandemic metrics and GTSVs.

**Table 1 ijerph-19-05613-t001:** Estimated keyword search volumes.

Variable	Keyword							Total
	Air Pollution	Smog	Air Quality	Air Quality Index	Air Purity	PM_10_	PM_2.5_	
Estimated number of searches in years 2016–2021	1,486,988	2,201,171	587,275	48,495	65,007	108,304	47,257	4,544,497
Relative weight	31.5	46.6	12.4	1.0	1.4	2.3	1.0	-

**Table 2 ijerph-19-05613-t002:** Comparison of the real-world pre- and intra-pandemic GTSVs.

Keyword	Mean GTSV Difference	Median GTSV Difference	Mean Percentage GTSV Difference	*p*-Value
Air pollution	−5.3	−1.5	−53.5	0.007
Smog	−3.3	−0.6	−51.0	0.036
Air quality	−0.9	4.0	−5.8	0.081
Air quality index	−0.8	1.1	−7.3	0.969
Air purity	−2.1	0.0	−20.4	0.244
PM_10_	0.4	−0.9	4.3	0.493
PM_2.5_	−0.6	3.7	−10.6	0.620
Arithmetic mean	−1.8	1.6	−18.7	0.577
Weighted mean	−3.5	0.0	−39.0	0.174

**Table 3 ijerph-19-05613-t003:** Comparison of the forecasted and real-world intra-pandemic GTSVs.

Keyword	SARIMA Hyperparameters	Mean Absolute Percentage Error [%]	Mean GTSV Difference	Median GTSV Difference	Mean GTSV Percentage Difference [%]	*p*-Value
Air pollution	(1,0,1) × (1,0,1)12	33.2	−2.4	−1.8	−33.0	0.509
Smog	(1,0,1) × (1,0,1)12	22.6	−0.9	−0.2	−19.7	0.907
Air quality	(3,1,3) × (2,1,3)12	6.5	−11.5	−14.7	−41.3	0.461
Air quality index	(1,0,1) × (3,0,1)12	2.2	−2.1	0.6	−16.3	0.954
Air purity	(1,1,1) × (1,1,2)12	17.6	−11.6	−14.9	−56.9	0.001
PM_10_	(3,0,1) × (1,0,1)12	44.3	0.7	1.1	7.8	0.081
PM_2.5_	(1,0,1)× (1,0,1)12	3.0	3.0	1.6	132.6	0.008
Arithmetic mean	(1,0,1) × (1,0,1)12	10.2	−0.3	0.5	−3.3	0.727
Weighted mean	(1,0,1) × (1,0,1)12	6.2	−1.2	−0.3	−16.5	0.969

## Data Availability

Publicly available datasets were analysed in this study. This data can be found here: 1. Google Trends: https://trends.google.pl/trends/?geo=PL (accessed on 30 January 2022); 2. Our World in Data: https://ourworldindata.org/coronavirus (accessed on 30 January 2022); 3. KWFinder: https://app.kwfinder.com/ (accessed on 30 January 2022).

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
