# Peer review of "Changes in Air-Pollution-Related Information-Seeking Behaviour during the COVID-19 Pandemic in Poland"

_ijerph, 2022, doi:10.3390/ijerph19095613_

Round 1

Reviewer 1 Report

It is an interesting work that touches upon important aspects for a country like Poland, where air pollution compared to the other EU Member States is still a huge problem. The figures are of good quality and very well presented. Statistical analyzes are performed correctly with all standards. The strength of the analysis is that the Authors used the objective criterion of information for hyperparameter tuning. English is at a good level and requires no corrections. The topic is interesting and important in Poland. The work is in line with the journal scope.

In my opinion, the work is ready for publication after introducing minor addition:

In the introduction and discussion, I recommend adding recent papers analyzing the change in air pollution trends (from sensors data) in southern Poland over the decades, but also the impact of the COVID-19 pandemic and related changes in population behavior. I wonder if the analysis of these studies somehow coincides with your conclusions (I am talking about potential people's behavioral mechanisms). Please look for papers about Poland in a special issue of Aerosol and Air Quality Research named Air Quality in a Changed World: Regional, Ambient, and Indoor Air Concentrations from the COVID to Post-COVID Era and in MDPI Sensors special issue Sensors for Air Quality Monitoring. 

Specific comments for authors

Not all need to be reflected in the text. 

Line 133 often people use PM2.5 and PM25 interchangeably, so it would be good to search for both them and combine them into one feature.

Line 139-142 - what is the uncertainty of such estimations?

Line 172 - did you tired also BIC? 

Line 340 - why improbable? maybe the fear of serious effects of the disease was the motivation to analyze other factors that may affect the respiratory system

Line 372 - 390 did you consider the "habituation effect"? The problem of air pollution in Poland has been publicized for a decade, maybe people simply do not look for more information because they know it.

Author Response

### Dear Reviewer, thank you for your comments and your positive opinion on our paper. We listed our answers to your comments below:

It is an interesting work that touches upon important aspects for a country like Poland, where air pollution compared to the other EU Member States is still a huge problem. The figures are of good quality and very well presented. Statistical analyzes are performed correctly with all standards. The strength of the analysis is that the Authors used the objective criterion of information for hyperparameter tuning. English is at a good level and requires no corrections. The topic is interesting and important in Poland. The work is in line with the journal scope.

 In my opinion, the work is ready for publication after introducing minor addition:

In the introduction and discussion, I recommend adding recent papers analyzing the change in air pollution trends (from sensors data) in southern Poland over the decades, but also the impact of the COVID-19 pandemic and related changes in population behavior. I wonder if the analysis of these studies somehow coincides with your conclusions (I am talking about potential people's behavioral mechanisms). Please look for papers about Poland in a special issue of Aerosol and Air Quality Research named Air Quality in a Changed World: Regional, Ambient, and Indoor Air Concentrations from the COVID to Post-COVID Era and in MDPI Sensors special issue Sensors for Air Quality Monitoring. 

We cited your papers in the introduction:

“Even though the air pollution levels in large polish urban areas over the last decade are decreasing, World Bank Group states that 36 of the 50 most polluted cities in the European Union are in Poland [2,3].”

And:

“On the other hand, during the lockdowns, such behavioural changes resulted in local air pollution reduction [24,25].”

As well as in the discussion:

“Fortunately, based on the example of Cracow, a large city located in southern Poland, it is clearly visible that education and multilevel political involvement are effective initiatives in increasing environmental awareness of the local population which results in a stable decrease in emissions [3]”

Specific comments for authors

Not all need to be reflected in the text. 

Line 133 often people use PM2.5 and PM25 interchangeably, so it would be good to search for both them and combine them into one feature.

###We agree with that statement. However, to conduct the same searches for all keywords, we decided to search for just one version of each keywords, the one, which we considered the most common one. However, in our further studies, according to your suggestion, we will combine keywords with the same semantic meaning but different spelling (like pm2.5 and pm25)

Line 139-142 - what is the uncertainty of such estimations?

###We added the sentence in the Section 4.3 Limitations:

The changes of keywords’ popularity might have also affected the absolute search volumes estimated with use of KWFinder. Therefore, the uncertainty of the calculated absolute search volumes may be high and is difficult to estimate it.

Line 172 - did you tired also BIC?

###We did not try the BIC, as in all cases the AIC and MAPE were in line with each other, and therefore we thought that two estimators of error of different types are sufficient to evaluate the SARIMA models.

Line 340 - why improbable? maybe the fear of serious effects of the disease was the motivation to analyze other factors that may affect the respiratory system

###We changed this part:

This suggests that an increase in COVID-19 cases and/or deaths would result in an in-crease in air pollution-related information-seeking behaviours, which is possible, as the fear of serious threat of the COVID-19 infection might have motivated people to analyse other factors that might have affected the respiratory system at that time.

On the other hand, if the correlations supported the observed decreases in mean GTSVs (Table 3), negative correlation coefficients would be expected. In addition to that, it was noted that the intensity of “COVID-19” keyword search queries was certainly not correlated with the changes in both forecasted and actual mean GTSVs (ρ from 0.055 to 0.11). Therefore, based on the data provided by the correlation coefficients, the changes in air pollution-related information-seeking behaviours seem to be independent of the events of the COVID-19 pandemic.”

Line 372 - 390 did you consider the "habituation effect"? The problem of air pollution in Poland has been publicized for a decade, maybe people simply do not look for more information because they know it.

###We agree. We added the following sentences in the last part of the 4.1 section:

Last but not least, the problem of low air quality in Poland is being measured for over 20 years [43]. As this issue was highly promoted in the last decade, habituation to air pollution by Poles might have occurred. Recently, similar observations were de-scribed for several highly polluted countries [44]. As suggested by the authors of this study, such behaviour may threaten the societies’ collaboration for air pollution reduction. This emphasizes the alarming evidence of our study and the constant need of local air pollution awareness promotion, especially in highly polluted regions of Poland.

Reviewer 2 Report

I read with great interest a study on changes in air pollution-related in formation-seeking behaviour during the COVID-19 pandemic in Poland. The article mainly has the following problems.

  1. There are problems with abbreviation, after abbreviation appears for the first time in the paper, they should be replaced in the following text. Such as Google Trends search volumes should be replaced by the abbreviation (GTSV) in the figure and table titles.
  2. In Figure 2(a)~(i), the identification position of the serial number of the figures is easy to cause misunderstanding. It is suggested to mark the series number to the upper left corner in the figures.
  3. The layout of the figures are not compact enough. The abscissa of Figure2(a)~(i) is consistent and it is suggested to use the same abscissa.

Author Response

### Dear Reviewer, thank you for your comments and your positive opinion on our paper. We listed our answers to your comments below:

Reviewer 2

  1. There are problems with abbreviation, after abbreviation appears for the first time in the paper, they should be replaced in the following text. Such as Google Trends search volumes should be replaced by the abbreviation (GTSV) in the figure and table titles.

We corrected this issue.

  1. In Figure 2(a)~(i), the identification position of the serial number of the figures is easy to cause misunderstanding. It is suggested to mark the series number to the upper left corner in the figures.

###We added the marks, as suggested.

  1. The layout of the figures are not compact enough. The abscissa of Figure2(a)~(i) is consistent and it is suggested to use the same abscissa.

###We corrected the figures.

Reviewer 3 Report

The authors present a study comparing pre- and intra-pandemic, real-world and modeled interest in keywords related to air pollution and COVID-19. This study is well-written with the exception of the first part of the Discussion section which is too long. Specific comments are listed below:

Line 17: Either remove the "-" or say "changed by -39.0%" since it is confusing to say "decreased by -39.0%". I leave this decision to the editor as it is used throughout the manuscription.

Line 47: Consider adding "(county)" or "(province)" since voivodeship is not a well-known word.

Line 50: The WHO updated "nitrogen oxide" (not "nitric oxide").

Lines 116-7: If the first mentions of the pandemic-related terms occurred in the week of 19/01/2020-25/01/2020, why was that week not included in the intra-pandemic time period?

Table 1: Please make sure to subscript "2.5" and "10" following "PM".

Figure 1: The figure is of very low resolution and should be remade. The y-axis should have "search volume" capitalized as the acronym is "GTSV".

Figure 2: The y-axis should have "search volume" capitalized as the acronym is "GTSV". The lines are difficult to tell apart - maybe one should be dashed or at least made thicker to be able to distinguish the green and red, in particular.

Line 254: It should be "Figure 2.f" when referring specifically to the PM10 figure.

Figure 4 is of low resolution and should be re-made.

Section 4: There is no Section 4.1. Also this first section is very long and should be subdivided for easier reading.

Lines 319-320: The authors note the irregular patterns of PM10 but do not offer an explanation. I am not very well aware of air pollution challenges in Poland, but is it possible that dust storms or other irregular/exceptional events are responsible for PM10 and its social awareness?

Line 326: Instead of "by -33.0%" use "of -33.0%".

Line 348: Is "greatest" supposed to mean "largest" or "most populated"?

Line 397: Instead of "changed also" say "also changed".

Figure 3: The figure is of very low resolution and should be remade. The y-axis should have "search volume" capitalized as the acronym is "GTSV".

Author Response

### Dear Reviewer, thank you for your comments and your positive opinion on our paper. We listed our answers to your comments below:

The authors present a study comparing pre- and intra-pandemic, real-world and modeled interest in keywords related to air pollution and COVID-19. This study is well-written with the exception of the first part of the Discussion section which is too long. Specific comments are listed below:

Line 17: Either remove the "-" or say "changed by -39.0%" since it is confusing to say "decreased by -39.0%". I leave this decision to the editor as it is used throughout the manuscription.

###We changed to “changed by -39.0%” version throughout the manuscript

Line 47: Consider adding "(county)" or "(province)" since voivodeship is not a well-known word.

###We added the word “province”.

Line 50: The WHO updated "nitrogen oxide" (not "nitric oxide").

###Corrected.

Lines 116-7: If the first mentions of the pandemic-related terms occurred in the week of 19/01/2020-25/01/2020, why was that week not included in the intra-pandemic time period?

###We wanted to include the first whole week of the pandemic. Therefore, we started from the 26/01/2020

Table 1: Please make sure to subscript "2.5" and "10" following "PM".

### Corrected

Figure 1: The figure is of very low resolution and should be remade. The y-axis should have "search volume" capitalized as the acronym is "GTSV".

### corrected. We changed “google trends search volume” to “GTSV”, as suggested by the 2nd reviewer.

Figure 2: The y-axis should have "search volume" capitalized as the acronym is "GTSV". The lines are difficult to tell apart - maybe one should be dashed or at least made thicker to be able to distinguish the green and red, in particular.

### corrected

Figure 3: The figure is of very low resolution and should be remade. The y-axis should have "search volume" capitalized as the acronym is "GTSV".

### corrected

Line 254: It should be "Figure 2.f" when referring specifically to the PM10 figure.

### corrected

Figure 4 is of low resolution and should be re-made.

### corrected

Section 4: There is no Section 4.1. Also this first section is very long and should be subdivided for easier reading.

### We changed the layout of the Section 4 (we added two additional subsections)

Lines 319-320: The authors note the irregular patterns of PM10 but do not offer an explanation. I am not very well aware of air pollution challenges in Poland, but is it possible that dust storms or other irregular/exceptional events are responsible for PM10 and its social awareness?

### We added a possible explanation”

“Most probably, causes of the poor fit of the “PM10” model are relatively low popularity of the keyword as well as possible interference of this search term with other longer or less precise search queries.”

Line 326: Instead of "by -33.0%" use "of -33.0%".

### corrected

Line 348: Is "greatest" supposed to mean "largest" or "most populated"?

### corrected to “largest”

Line 397: Instead of "changed also" say "also changed".

### corrected

Round 2

Reviewer 3 Report

The authors addressed my concerns and comments thoroughly.